# Chain of Visual Perception: Harnessing Multimodal Large Language Models for Zero-shot Camouflaged Object Detection

## ABSTRACT

In this paper, we introduce a novel multimodal camo-perceptive framework (MMCPF) aimed at handling zero-shot Camouflaged Object Detection (COD) by leveraging the powerful capabilities of Multimodal Large Language Models (MLLMs). Recognizing the inherent limitations of current COD methodologies, which predominantly rely on supervised learning models demanding extensive and accurately annotated datasets, resulting in weak generalization, our research proposes a zero-shot MMCPF that circumvents these challenges. Although MLLMs hold significant potential for broad applications, their effectiveness in COD is hindered and they would make misinterpretations of camouflaged objects. To address this challenge, we further propose a strategic enhancement called the Chain of Visual Perception (CoVP), which significantly improves the perceptual capabilities of MLLMs in camouflaged scenes by leveraging both linguistic and visual cues more effectively. We validate the effectiveness of MMCPF on four widely used COD datasets, containing CAMO, COD10K, NC4K and MoCA-Mask. Experiments show that MMCPF can outperform all existing state-of-the-art zero-shot COD methods, and achieve competitive performance compared to weakly-supervised and fully-supervised methods, which demonstrates the potential of our proposed MMCPF.

## CCS CONCEPTS

• **Computing methodologies** → **Scene understanding**.

## KEYWORDS

Zero-shot Camouflaged Object Detection, Multimodal Large Language Models, Chain of Visual Perception.

## 1 INTRODUCTION

Aimed at accurately identifying objects that blend seamlessly into their surroundings, Camouflaged Object Detection (COD) focuses on detecting those that are deliberately disguised. Yet, the field is hampered by a reliance on supervised learning models [1, 10, 11, 15, 19, 34, 36, 39, 47–49, 51, 56, 58], which demand extensive, accurately annotated COD datasets. However, camouflaged objects can be diverse, ranging from animals in various environments to a variety of man-made items. Some camouflaged objects may be rare or difficult to gather sufficient labeled data for training. In practical applications, new types of camouflage may continually emerge. Therefore, training models solely on fixed, small-scale datasets may limit their generalizability, rendering them effective in certain scenarios but ineffective in others. Specifically, as shown in Table. 1, when transferring the typical fully-supervised COD method ERRNet [18] to the new camouflaged scene, such as video camouflaged object detection (VCOD) scene, their performance significantly decreases, even falling below that of the weakly-supervised COD method WSCOD [13] and the zero-shot VCOD method MG [46]. Therefore,

**Table 1: Comparison of MMCPF and other methods. CAMO [25] and MoCA-Mask [5] are two different datasets containing different camouflaged scenes. The method ERR-Net may only achieve good performance in one certain scene (CAMO) while failing in another scene (MoCA-Mask).**

| Methods | Setting | CAMO | MoCA-Mask |
|---|---|---|---|
| | | $F_\beta^w$ | $F_\beta^w$ |
| MG (ICCV2021) | Zero-shot | - | 0.168 |
| ERRNet (PR2022) | Fully-supervised | 0.679 | 0.094 |
| WSCOD (AAAI2023) | Weakly-supervised | 0.641 | 0.121 |
| Ours | Zero-shot | 0.680 | 0.196 |

how to design a novel COD framework in the zero-shot manner with potential for generalizability is a matter worth exploring.

The first reason with the limited generalizability of current COD network designs arises from the small scale of training data, which tends to lead the networks to overfit in specific scenarios. This overfitting restricts their ability to generalize effectively in new environments. Additionally, existing COD network designs stem from their limited model capacity, which restricts their ability to perceive camouflaged objects across varied environments. In recent years, natural language processing (NLP) has been profoundly transformed by the advent of large language models (LLMs). These foundational models have demonstrated exceptional generalizability because they possess strong model capacity and millions or even tens of millions of training data. The fusion of LLMs with vision systems has led to the emergence of Multimodal Large Language Models (MLLMs) [4, 6, 16, 26, 54, 55, 57], such as LLaVA [31] and GPT-4V [37]. Leveraging the generalization capabilities of MLLMs, many researchers have explored and designed various zero-shot frameworks to address diverse visual tasks [24, 32, 40, 41, 53].

Therefore, in this paper, building upon the generalizability of MLLMs in different scenarios, we are inspired to explore whether MLLMs can maintain their efficacy in COD scenes and design a novel MLLM-based zero-shot COD framework, namely multimodal camo-perceptive framework (MMCPF). It is imperative to underscore that the primary objective of this investigation is not to retrain MLLMs using adapters or tuning mechanisms [35, 42, 50] on the small-scale COD dataset, thereby compromising the generalizability inherent in MLLMs [20]. What we aim to explore is whether it's possible to design a novel zero-shot framework that directly leverages the inherent model capacity of MLLMs to perceive camouflage scenarios without the annotated COD training dataset.

The development of zero-shot MMCPF faces two significant challenges: The first is how to perceive and locate camouflaged objects, and the second is how to generate the binary mask of camouflaged objects. To preliminarily address these challenges, MMCPF consists of two foundational models: one is MLLM, which is responsible for perceiving and locating the coordinates of the

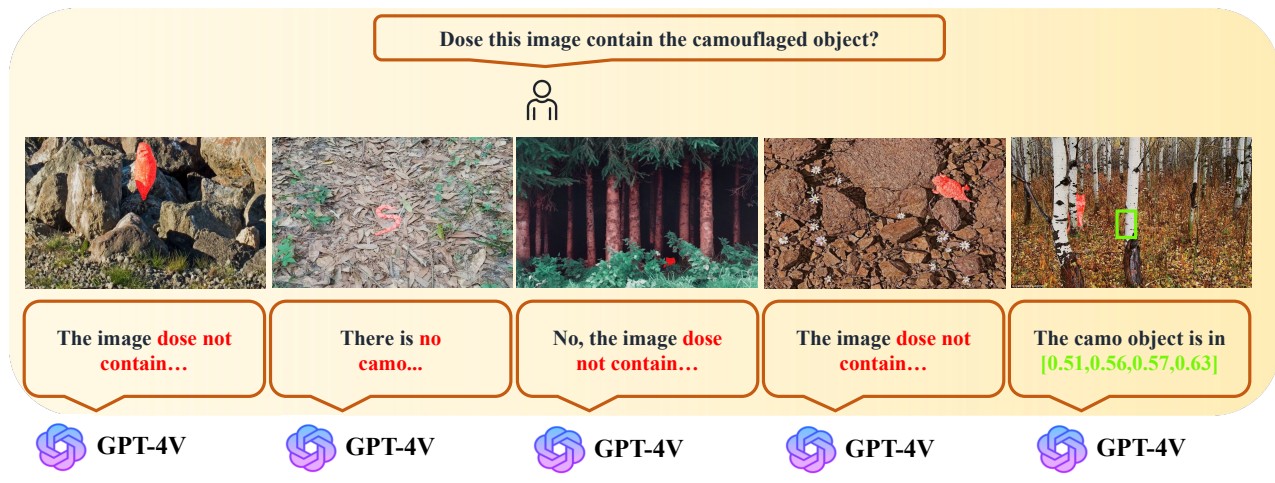

**Figure 1: Querying results generated by GPT-4V in COD. GPT-4V would answer the question incorrectly or randomly guess some wrong answers. The red mask is generated by ground-truth and The green box is generated by GPT-4V.**

camouflaged object. The other is a promptable visual foundation model (VFM) like SAM [21], which takes the coordinates outputted by MLLM to generate a binary mask. However, when we attempt to ask GPT-4V [37] about the presence of a camouflaged object in Fig. 1, we regret to find that the MLLM outputs some wrong contents. Therefore, a question raised: *Can even a powerful model like GPT-4V not effectively handle camouflaged scenes? How can we make zero-shot MMCPF work?*

Upon identifying the aforementioned issues, we design the chain of visual perception (CoVP) within MMCPF, which can not only help enhance MLLM's perception of camouflaged scenes and correctly output location coordinates, but also improve these coordinates to prompt VFM for mask generation. Specifically, our proposed CoVP improves the performance of MMCPF from both linguistic and visual perspectives. For linguistic perspective, inspired by Chain of Thought (CoT) techniques [12, 45] used in LLMs, we have enhanced the perceptual abilities of MLLMs for camouflaged scenes. In general, CoT can effectively help LLMs solve some complex downstream tasks under the zero-shot manner. However, how to design these reasoning mechanisms effectively for MLLMs remains an area to be explored. The work [3] attempts to facilitate visual reasoning in MLLMs by artificially providing semantic information about a given image in the text prompt. But in the COD task, the semantic and location information of the camouflaged object needs to be perceived and discovered by the model itself, rather than being artificially provided. That is to say, the method presented in paper [3] is not directly applicable to MMCPF. Therefore, in this paper, we summarize three critical aspects when prompting the MLLM to perceive the relationship between the camouflaged object and its surroundings, thereby enhancing the accuracy of MLLMs in locating camouflaged objects.

For visual perspective, considering the fact that the integration of additional visual information into MLLMs introduces unique challenges not encountered with text-only LLMs. These challenges are particularly pronounced in visually complex scenarios involving camouflage. While we have implemented a reasoning mechanism

within the text input to assist MLLMs in recognizing camouflaged objects, this does not entirely assure the accuracy of their visual localization capabilities. Therefore, in MMCPF, it is crucial to consider how to improve the uncertain pixel location from MLLMs to better prompt VFMs to generate accurate binary mask results. To address this challenge, we design a mechanism called visual completion. As illustrated in Table. 1, with the help of CoVP, MMCPF outperforms those published in 2023 with weakly-supervised settings, demonstrating remarkable potential for a zero-shot framework. Notably, MMCPF also surpasses fully-supervised methods published in 2022.

In summary, the key contributions are listed as follows:

- We propose MMCPF to preliminarily explore the performance cap of MLLMs to camouflaged scenes in the zero-shot manner. We hope MMCPF can potentially inspire researchers to design COD framework from a fresh perspective without the needing of training process and annotated COD training datasets, which is more flexible.
- We introduce CoVP in MMCPF, which enhance MLLM's perception of camouflaged scenes from both linguistic and visual perspectives and make MMCPF work.
- We validate the proposed MMCPF on three widely used COD datasets CAMO [25], COD10K [11] and NC4K [34], and one VCOD dataset MoCA-Mask [5]. Its effective zero-shot performance verifies the effectiveness of our ideas.

## 2 RELATED WORK

### 2.1 Multimodal Large Language Model

Prompted by the powerful generalized ability of LLMs [2, 7, 33, 52] in NLP, VFMs [22, 29, 38, 43] have emerged. The integration of LLMs and VFMs has facilitated the advancement of MLLMs [4, 6, 16, 26, 28, 31, 37, 54, 55, 57]. MLLMs show case impressive visual understanding through end-to-end training techniques that directly decode visual and text tokens in a unified manner. These foundational models, like GPT, SAM and LLaVA, demonstrated the immense potential of these large-scale, versatile models, trained on

extensive datasets to achieve unparalleled adaptability across a wide range of tasks. This paradigm shift, characterized by significant strides in representation learning, has spurred the exploration of task-agnostic models, propelling research into both their adaptation methodologies and the intricacies of their internal mechanisms.

In the field of NLP, to migrate LLMs to downstream tasks without impacting the LLM's inherent performance, In-context learning [8] is a widely used technique. A particularly influential approach within In-context learning is the CoT [12, 23, 45]. CoT, by designing a series of reasoning steps, guides the LLM to focus on specific content at each step, thereby further stimulating the model's innate logical reasoning capabilities. Specifically, these works have discovered that by prompting LLMs with designed directives, such as *"let's think step by step"*, the reasoning abilities of the LLMs can be further improved and enhanced.

As a field that is more mature than MLLM, LLM has shown that models can be effectively migrated to various downstream tasks in a zero-shot manner, provided that the correct prompting mechanisms are in place. Therefore, to further advance the development of MLLM, our paper explores the upper limits of MLLM performance in visually challenging tasks, specifically COD. Unlike existing methods that use re-training or tuning to migrate MLLMs to downstream tasks [24, 40, 41], our paper aims to explore how to prompt the MLLMs to stimulate its inherent perception abilities for camouflaged scenes. To this end, we propose CoVP, which first identifies key aspects to consider when inputting language text prompts to enhance MLLM's understanding of camouflaged scenes. Further, we highlight how to utilize MLLM's uncertain visual outputs and, from a visual completion standpoint, enhance MLLM's capability to capture camouflaged objects.

## 2.2 Camouflaged Object Detection

In the past years, there has been significant effort in the COD task [1, 10, 11, 15, 19, 34, 36, 39, 47, 48, 51, 56, 58]. The technical frameworks for these COD methods can be categorized into two types: CNN-based and Transformer-based approaches. Although the structures of these methods may differ, their core lies in designing advanced network modules or architectures capable of exploring discriminative features. While these methods have achieved impressive performance, the networks lack generality and are task-specific, which limits their generalizability. This means that while they are highly effective for specific tasks, their adaptability to a wide range of different tasks is constrained.

The emergence of a series of foundational models in recent times has signaled to computer vision researchers that it is possible to solve a variety of downstream vision tasks using a single, large-scale model. This trend highlights the potential of leveraging powerful, versatile models that have been trained on extensive datasets, enabling them to handle diverse and complex visual challenges. In line with the trend of technological advancement, this paper explores the generalization capabilities of visual foundational models. We design the MMCPF to generalize the foundational model to the COD task in a zero-shot manner. It's important to highlight that this paper does not employ methods such as re-training, adapters, or tuning to update the parameters of the visual foundational model for adaptation to the COD task. Instead, we explore how to enhance the perception abilities of the visual foundational model in camouflaged scenes through prompt engineering from both linguistic and visual perspectives, without altering its inherent capabilities.

Existing works like ZSCOD [27] and GenSAM [14] are also attempting to design a zero-shot COD framework. The pipline of ZSCOD may seem somewhat contrived, potentially limiting its adaptability to real-world scenarios. Specifically, this method artificially divides the COD10K dataset into seen and unseen categories and then trains the network. This undoubtedly leads to the risk of information leakage, thus preventing a true evaluation of whether the network has generalizability. Moreover, its performance to unknown scenarios is very weak, raising concerns among researchers about the robustness of their networks. GenSAM [14] also leverages MLLM and VFM to design a zero-shot COD framework. However, a flaw of this framework is that it makes additional modifications to the foundation model (Clip-Surgery [30]) used in the framework. Specifically, to enhance the network's ability to filter noise from camouflaged images and accurately identify foreground pixels, Gen-SAM introduces an additional attention mechanism to the existing well-established framework. This approach is somewhat similar to an Adapter operation. This could potentially affect the generalizability that the foundation model originally had to some extent. Compared to these two methods, our MMCPF is more flexible in its setup. In MMCPF, we do not make any structural modifications to the MLLM and VFM. Instead, we enhance the MLLM's perceptual capabilities for camouflaged scenes through the design of the CoVP. This approach ensures the maximal preservation of the MLLM's inherent generalization ability, and achieve superior performance.

## 3 METHOD

### 3.1 Framework Overview

In Fig. 2, our proposed MMCPF is a promptable framework, which is the first framework to successfully generalize the MLLM to the camouflaged scene. Given an image $I$ containing the camouflaged scene, prompting the MMCPF with the text $\mathcal{T}$, such as *"Please find a camouflaged object in this image and provide me with its exact location coordinates"*, the MMCPF would locate the camouflaged object at position $\mathcal{P}_I$ and generate the corresponding mask $\mathcal{M}$.

To achieve the above purpose, we state that MMCPF needs two foundational models. The first is the MLLM, which can accept the user instruction and output the corresponding result, such as the coordinate information of the target object. To further quantitatively evaluate the performance of MLLM, the second foundational model is the promptable VFM, which can accept the output from MLLM as the prompt, and generate the final mask $\mathcal{M}$. Note that, in MMCPF, both MLLM and VFM are frozen.

In MMCPF, only having MLLM and VFM, despite their powerful capabilities, may still be insufficient to handle the COD task effectively. Specifically, in Fig. 2, using just **vanilla text prompts** to query the MLLM might yield meaningless results, contributing nothing to the accurate location of camouflaged objects. Additionally, in Fig. 3, the positional coordinates output by the MLLM may carry a degree of uncertainty, encompassing only a part of the camouflaged object. Specifically, for the localization of camouflaged objects, MLLM typically outputs the coordinates of the top-left and bottom-right corners. We have observed that these coordinates do

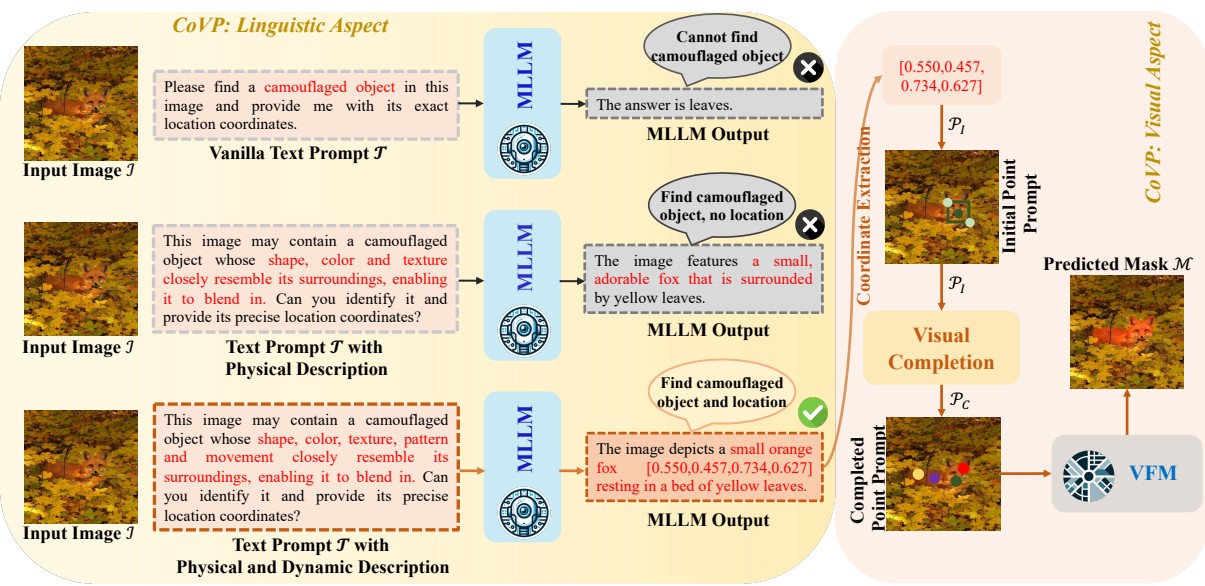

**Figure 2: Our multimodal camo-perceptive framework (MMCPF). MMCPF mainly contains chain of visual perception (CoVP), which is to enhance the perceptual abilities of the MLLM in camouflage scenarios from linguistic aspect and visual aspect.**

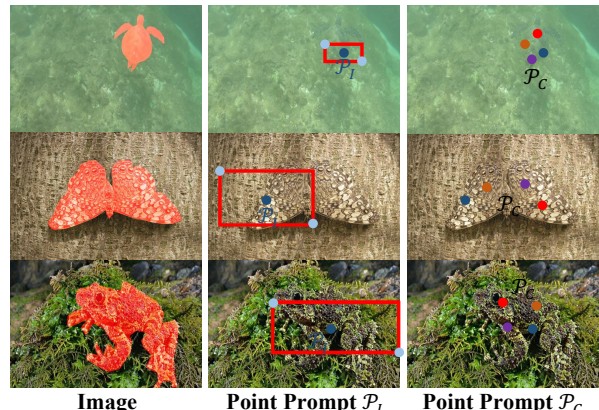

**Figure 3: Second column visualizes coordinates generated by MLLM, which are somewhat uncertain and cannot completely locate the camouflaged object. Third column displays coordinates generated by our visual completion mechanism. $\mathcal{P}_I$ and $\mathcal{P}_C$ are initial and completed points respectively.**

not always fall within the interior of the camouflaged object and their central point only lands on the edge of the camouflaged object sometimes. Consequently, if these coordinates are directly used as point prompts for the VFM, the resulting mask might be incomplete or fragmented. Therefore, to address the above problems, we propose CoVP, which enhance MLLM's perception of camouflaged scenes from both linguistic and visual perspectives.

## 3.2 Chain of Visual Perception

Images in camouflaged scenes obviously present visual challenges, making it difficult for MLLMs to detect camouflaged objects. The challenges primarily encompass two aspects. Firstly, for MLLM, we

stimulate its understanding of visual content in an image through language. However, designing language prompts suitable for camouflaged scenes remains an area to be explored. The existing work [3] attempts to enhance the visual perception capabilities of MLLM by providing semantic information about an image, which contradicts the definition of the COD task, and thus cannot be directly applied. Therefore, our first task is to design how language can be used to enhance the visual perception ability of MLLM.

Secondly, prompting MLLM to visually perceive an image through language represents a challenging cross-modal task, especially when we attempt to generalize MLLM to visually the challenging COD scene. As visualized in Fig. 3, it's difficult to completely ensure the accuracy of MLLM's output. Consequently, we design a visual completion (VC) to further enhance the localization capability of MLLM. Unlike the CoT, which only designs mechanisms at the text input of the LLM to enhance its language reasoning ability, CoVP attempts to improve MLLM's perception of camouflaged scenes more comprehensively, working at both the input and output and from linguistic and visual perspectives.

### 3.2.1 Perception Enhanced from Linguistic Aspect.

We attempt to design an effective text prompt mechanism from three perspectives to further enhance the ability of MLLM to perceive camouflaged objects. This primarily includes the following aspects: description of the attributes of the target camouflaged object, the angle of polysemy, and the perspective of diversity.
**Description of the attribute.** When prompting the MLLM to discover a specific camouflaged object, we should encourage the MLLM to pay attention to the potential attributes of that object. This includes two perspectives: internal attributes and external interaction. For the internal attributes of camouflaged objects, we

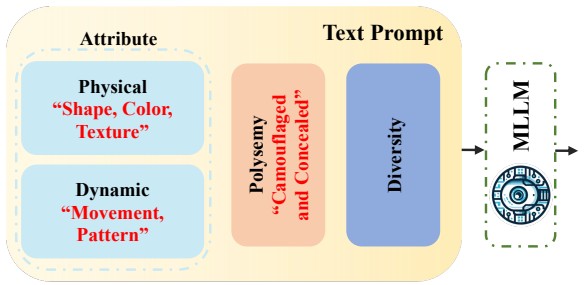

**Figure 4: Prompts with attribute, polysemy and diversity.**

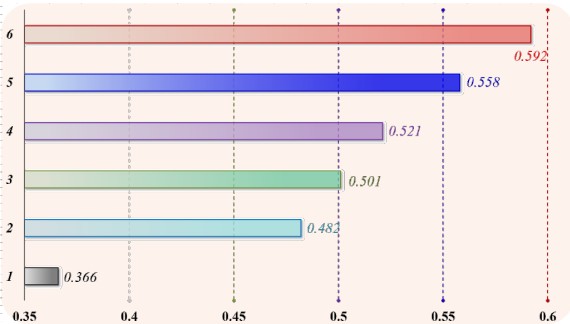

**Figure 5: Performance improvement in COD10K when adding 2.physical attribute description, 3.dynamic attribute description, 4.polysemous description, 5.diverse description and 6.visual completion compared to 1.baseline.**

aim to focus the MLLM on their physical and dynamic characteristics. Physical properties might include the camouflaged object's color, shape, and texture information, which are static attributes. For example, as shown in Fig. 4 and Fig. 5, when we try to include descriptions of these aspects, we find that the MLLM's ability to perceive camouflaged objects is significantly enhanced.

Dynamic characteristics include the camouflaged object's patterns and motion information, which might also cause it to blend with its surrounding environment. In Fig. 5, when we attempt to direct the MLLM's attention to descriptions of these dynamic aspects, its ability to perceive camouflaged objects is further enhanced. It's important to note that our text prompts do not explicitly give away information about the camouflaged object. For example, we do not use prompts like *"The camouflaged object in the image is an orange fox."* Instead, our prompts are designed to subtly guide the MLLM in identifying and understanding the characteristics of the camouflaged object without directly revealing it.

**Polysemy of the description.** It's important to consider polysemy when designing prompts. For example, the term "camouflage" can have different interpretations sometimes, where it can also refer to a soldier wearing camouflage clothing. Therefore, we would also design the text prompt such as *"This image may contain a concealed object..."*. In Fig. 5, when we design text prompts with consideration for polysemy, the ability to perceive camouflaged objects is improved. This observation underscores the importance of crafting prompts that account for different meanings and interpretations, thereby enabling the MLLM to more effectively process and understand the complexities inherent in camouflaged scenes.

**Diversity of the description.** It's essential to focus on the diversity of prompts. Given the uncertainty about which type of prompt is most suitable for an MLLM, prompts should be as varied as possible. Moreover, in maintaining diversity, we suggest leveraging the LLM itself to generate prompts with similar meanings. This approach ensures that the prompt texts are as close as possible to the data distribution that the MLLM can effectively process. In Fig. 5, when we take into account the diversity of text prompts, the ability to perceive camouflaged objects is further enhanced. This improvement suggests that incorporating a variety of prompts, which cover different aspects and perspectives, can significantly aid the MLLM in more effectively detecting camouflaged objects.

*3.2.2 Perception Enhanced from Visual Aspect.*

Through the text prompt we designed, we significantly enhance the MLLM's visual perception ability in challenging camouflaged

scenes, enabling us to preliminarily identify the location of camouflaged objects. However, it's important to note that the MLLM is initially designed for understanding image content, not for highly precise object localization. As a result, the MLLM's positioning of camouflaged objects is generally approximate and fraught with uncertainty. This is evident in Fig. 3, where the visualization of the MLLM's positioning results shows its limitations in accurately locating the entire camouflaged object. Using the MLLM's output coordinates as direct point prompts for the VFM in segmentation often leads to incomplete results. To tackle this challenge, we explore a solution: enhancing the initial, uncertain coordinates provided by the MLLM to improve its localization accuracy.

In Fig. 3, our goal is to generate additional points similar to the initial central point coordinate $\mathcal{P}_I$ in terms of semantics. Prior studies [38, 44] have shown that self-supervised vision transformer features, such as those from DINOv2, hold explicit information beneficial for semantic segmentation and are effective as KNN classifiers. DINOv2 particularly excels in accurately extracting the semantic content from each image. Hence, we utilize the features extracted by the foundational model DINOv2 to represent the semantic information of each image, denoted as $\mathcal{F}$. This approach enables us to more precisely expand upon the initial point coordinates, leveraging the semantic richness of DINOv2's feature extraction capabilities.

After generating the feature representation $\mathcal{F}$ of the input image $\mathcal{I}$, we obtain the feature vector $\mathcal{F}_I$ corresponding to the point $\mathcal{P}_I$. We then facilitate interaction between the feature vector $\mathcal{F}_I$ and other point features in $\mathcal{F}$ to calculate their correlation matrix. Specifically, in the image feature $\mathcal{F}$, which contains $N$ pixels, the feature representation of each pixel is denoted as $\mathcal{F}_C^i$, where $i \in [1, N]$. The correlation score between $\mathcal{F}_I$ and $\mathcal{F}_C^i$ is determined using cosine similarity. Subsequently, we employ a Top-k algorithm to identify the points most semantically similar to $\mathcal{F}_I$. These points are located at position $P$:

$$Sim = \mathcal{F}_C \times \mathcal{F}_I, P = \text{Top-k}(Sim) \in \mathbb{R}^K, \quad (1)$$

where $\times$ means matrix multiplication. Finally, we further refine the $P$ into $c$ clustering centers as the positive point prompts $\mathcal{P}_C$ for VFM. The point prompts $\mathcal{P}_C$ and the image $\mathcal{I}$ are sent to VFM to predict segmentation results $\mathcal{M}$.

Table 2: Comparison of MMCPF and other methods. "F" means fully-supervised methods. "ZS" means zero-shot methods. "WS" means weakly-supervised methods. Red font represent the top performance under the ZS setting. Gray Background indicates the metrics fully-supervised and weakly-supervised approaches under-perform MMCPF.

| Methods | Setting | CAMO (250 Images) | | | COD10K (2026 Images) | | | NC4K (4121 Images) | | |
|---------|---------|-----------|-------|------|-----------|-------|------|-----------|-------|------|
| | | $F_\beta^w$ | $S_\alpha$ | MAE | $F_\beta^w$ | $S_\alpha$ | MAE | $F_\beta^w$ | $S_\alpha$ | MAE |
| FSPNet (CVPR2023) | F | 0.799 | 0.856 | 0.050 | 0.735 | 0.851 | 0.026 | 0.816 | 0.879 | 0.035 |
| NCHIT (CVIU2022) | F | 0.652 | 0.784 | 0.088 | 0.591 | 0.792 | 0.049 | 0.710 | 0.830 | 0.058 |
| ERRNet (PR2022) | F | 0.679 | 0.779 | 0.085 | 0.630 | 0.786 | 0.043 | 0.737 | 0.827 | 0.054 |
| WSCOD (AAAI2023) | WS | 0.641 | 0.735 | 0.092 | 0.576 | 0.732 | 0.049 | 0.676 | 0.766 | 0.063 |
| ZSCOD (TIP2023) | ZS | * | * | * | 0.144 | 0.450 | 0.191 | * | * | * |
| GenSAM (AAAI2024) | ZS | 0.655 | 0.730 | 0.117 | 0.584 | 0.731 | 0.069 | 0.665 | 0.754 | 0.087 |
| Ours (Shikra+SAMHQ) | ZS | 0.680 | 0.749 | 0.100 | 0.592 | 0.733 | 0.065 | 0.681 | 0.768 | 0.082 |
| Ours (LLaVA+SAMHQ) | ZS | 0.677 | 0.747 | 0.103 | 0.595 | 0.734 | 0.067 | 0.682 | 0.766 | 0.081 |
| Ours (Shikra+SAM) | ZS | 0.677 | 0.747 | 0.103 | 0.589 | 0.732 | 0.069 | 0.677 | 0.766 | 0.087 |
| Ours (LLaVA+SAM) | ZS | 0.675 | 0.751 | 0.102 | 0.590 | 0.735 | 0.066 | 0.678 | 0.769 | 0.084 |

Table 3: Comparison of MMCPF and other methods in the VCOD scene. Abbreviations and symbols have the same meaning as in Table. 2.

| Methods | Setting | MoCA-Mask | | |
|---------|---------|-----------|-------|------|
| | | $F_\beta^w$ | $S_\alpha$ | MAE |
| SLTNet (CVPR2022) | F | 0.292 | 0.628 | 0.034 |
| PFSNet (CVPR2023) | F | 0.060 | 0.508 | 0.017 |
| ERRNet (PR2022) | F | 0.094 | 0.531 | 0.052 |
| WSCOD (AAAI2023) | WS | 0.121 | 0.538 | 0.041 |
| MG (ICCV2021) | ZS | 0.168 | 0.530 | 0.067 |
| GenSAM (AAAI2024) | ZS | 0.141 | 0.523 | 0.070 |
| Ours (Shikra+SAMHQ) | ZS | 0.196 | 0.569 | 0.031 |
| Ours (LLaVA+SAMHQ) | ZS | 0.192 | 0.571 | 0.030 |
| Ours (Shikra+SAM) | ZS | 0.188 | 0.560 | 0.034 |
| Ours (LLaVA+SAM) | ZS | 0.189 | 0.562 | 0.035 |

# 4 EXPERIMENTS

## 4.1 Datasets and Evaluation Metrics

We employ three public benchmark COD datasets to evaluate the perceptual capabilities of MMCPF in camouflaged scenes. These datasets include CAMO [25], COD10K [11] and NC4K [34]. CAMO is a subset of the CAMO-COCO, specifically designed for camouflaged object segmentation. COD10K are collected from various photography websites and are classified into 5 super-classes and 69 sub-classes. NC4K features more complex scenarios and a wider range of camouflaged objects. *To further demonstrate the generalization capability of our proposed MMCPF, we extend our validation to another dataset which may contain different camouflaged scene, MoCA-Mask [5], to assess our method's performance.* We adopt three widely used metrics to evaluate our method: structure-measure ($S_\alpha$) [9], weighted F-measure ($F_\beta^w$), and mean absolute error (MAE).

## 4.2 Implementation Details

To ensure the reproducibility of our MMCPF, thereby positively impacting the community, we select open-source models for both the MLLM and VFM. For the MLLM, we choose Shikra-7B [4] and LLaVA1.5-7B [31]. We do not opt for the potentially more powerful GPT-4V [37] as it is not open-source, and thus, its use would not guarantee the reproducibility of our framework. For the VFM, we selecte the SAM-HQ [20] and SAM [21].

## 4.3 Comparison COD and VCOD Methods

Selecting appropriate comparison methods is crucial to demonstrate the contribution of our proposed MMCPF to the community. The core of our MMCPF is to generalize MLLM and VFM to camouflaged scenes in a zero-shot manner. Since the MLLM and VFM we chose are not specifically designed for camouflaged scenes, we first compare our method with the zero-shot COD method ZSCOD [27] and GenSAM [14][1]. Secondly, as we do not retrain the MLLM and VFM on camouflaged datasets when generalizing them to camouflaged scenes, it is appropriate to compare our approach with unsupervised COD methods. Unfortunately, we could not find any unsupervised methods specifically designed for the COD task, so we opt for a comparison with the weakly-supervised method WSCOD [13]. Finally, we also compare our method with three fully-supervised approaches, including NCHIT [49], ERRNet [18], and FSPNet [17]. This comparison not only helps researchers understand the performance level of our paper but also further clarifies our contribution to the field. By setting our work against a backdrop of various supervisory approaches, we provide a comprehensive view of where MMCPF stands in the context of current COD methodologies and highlight its potential advantages. In addition to comparisons with current state-of-the-art COD methods, we also benchmark against two VCOD methods, SLTNet [5] and MG [46], which can demonstrate the generalization capability of MMCPF.

## 4.4 Quantitative and Qualitative Evaluation

**COD Evaluation.** From Table. 2, it is evident that the performance of our MMCPF significantly surpasses the zero-shot method ZS-COD [27]. This observation preliminarily reflects the generalization capabilities of MLLM in camouflaged scenes. Compared to another zero-shot method GenSAM [14], based on MLLM/VFM,

---

[1]The primary reason for the discrepancy in GenSAM's performance compared to the original paper is that we are unable to directly run the code provided by GenSAM. We make some debugging and modifications based on their code.

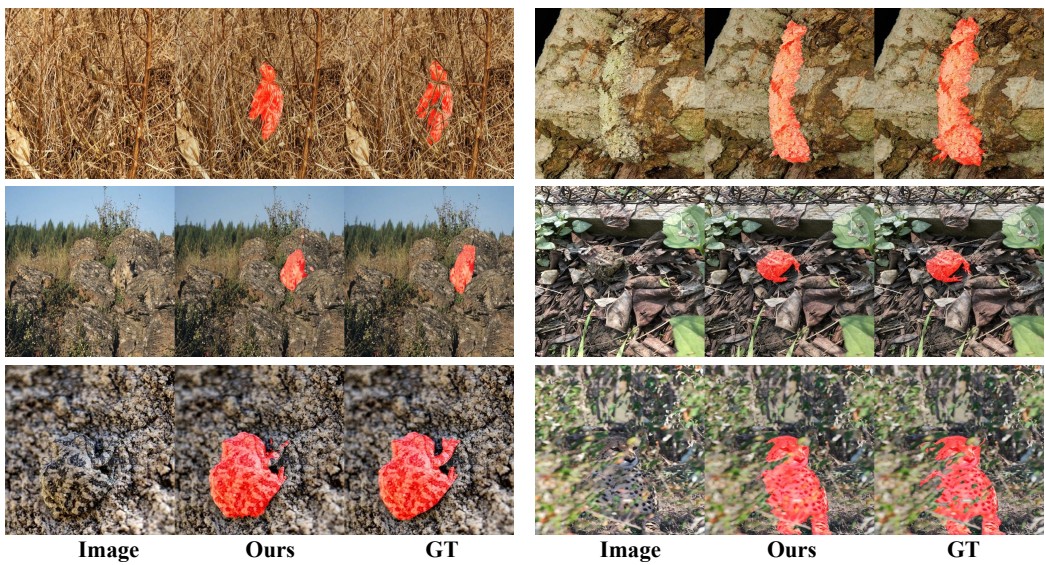

**Figure 6: Qualitative results of the proposed MMCPF framework.**

**Table 4: Ablation studies of MMCPF. PA means physical attribute. DA means dynamic attribute. VC means visual completion.**

| Methods | CAMO (250 Images) | | | COD10K (2026 Images) | | | NC4K (4121 Images) | | |
|---|---|---|---|---|---|---|---|---|---|
| | $F_\beta^w \uparrow$ | $S_\alpha \uparrow$ | MAE $\downarrow$ | $F_\beta^w \uparrow$ | $S_\alpha \uparrow$ | MAE $\downarrow$ | $F_\beta^w \uparrow$ | $S_\alpha \uparrow$ | MAE $\downarrow$ |
| 1. Baseline | 0.410 | 0.519 | 0.199 | 0.366 | 0.507 | 0.188 | 0.402 | 0.520 | 0.185 |
| 2. Baseline+PA | 0.554 | 0.629 | 0.157 | 0.482 | 0.615 | 0.127 | 0.565 | 0.651 | 0.143 |
| 3. Baseline+PA+DA | 0.573 | 0.649 | 0.149 | 0.501 | 0.640 | 0.120 | 0.580 | 0.681 | 0.126 |
| 4. Baseline+PA+DA+Polysemy | 0.603 | 0.671 | 0.134 | 0.521 | 0.663 | 0.107 | 0.605 | 0.701 | 0.121 |
| 5. Baseline+PA+DA+Polysemy+Diverse | 0.635 | 0.707 | 0.118 | 0.558 | 0.701 | 0.081 | 0.639 | 0.737 | 0.105 |
| 6. Baseline+PA+DA+Polysemy+Diverse+VC | 0.680 | 0.749 | 0.100 | 0.592 | 0.733 | 0.065 | 0.681 | 0.768 | 0.082 |

our method's performance can still surpass this method. A major reason for this is the unique nature of camouflaged objects. Although the GenSAM also designes some text prompts to guide the MLLM to detect camouflaged objects, the effectiveness of these prompts is not as comprehensive as the approach summarized in our paper, which prompts the MLLM from three distinct dimensions to identify camouflaged objects. Note that, under our framework, replacing the MLLM/VFM with any other does not significantly affect the performance of MMCPF. This adaptability to different MLLMs/VFMs without a significant performance drop highlights the flexibility and future potential of MMCPF. Furthermore, MM-CPF outperforms the weakly-supervised method WSCOD [13] in terms of $F_\beta^w$ and $S_\alpha$, an undoubtedly exciting performance indication. This suggests that with the design of appropriate enhancement mechanisms, MLLM can effectively perceive camouflaged objects. Finally, on the CAMO and COD10K datasets, the $F_\beta^w$ metric even surpasses some fully-supervised methods. This demonstrates the superiority of MMCPF in the localization capability of camouflaged objects. However, when compared with the current state-of-the-art fully-supervised method like FSPNet [17], there is still a noticeable performance gap. Also, the shortcomings in the MAE metric indicate that there is room for improvement in the absolute accuracy

of pixel-level predictions by MLLM/VFM, perhaps due to the lack of specific optimizations for downstream segmentation tasks in these foundational models. The visualization results in the Fig. 6 also show that MMCPF can better locate the camouflaged objects in different camouflage scenarios.

**VCOD Evaluation.** As shown in Table. 3, MMCPF not only completely surpasses the weakly-supervised method and unsupervised video COD methods, but it also achieves competitive results compared to some fully-supervised methods. This performance on the video data further underscores the versatility and adaptability of our approach. We can further find that GenSAM has limited generalizability on the new camouflaged scene. The primary reason causing this is that GenSAM modifies the structure of the original foundational model, which is like Adapter operation and might undermine the generalizability of the foundational model itself.

## 4.5 Ablation Studies

Herein, we use the ***Shikra + SAMHQ*** setting to conduct the ablation studies on CAMO, COD10K and NC4K datasets.

**Main Component.** In Table. 4, Baseline represents our use of the vanilla text prompt, *"Please find a camouflaged object in this image and provide me with its exact location coordinates"*, to query MLLM, without incorporating visual completion. The results in the first

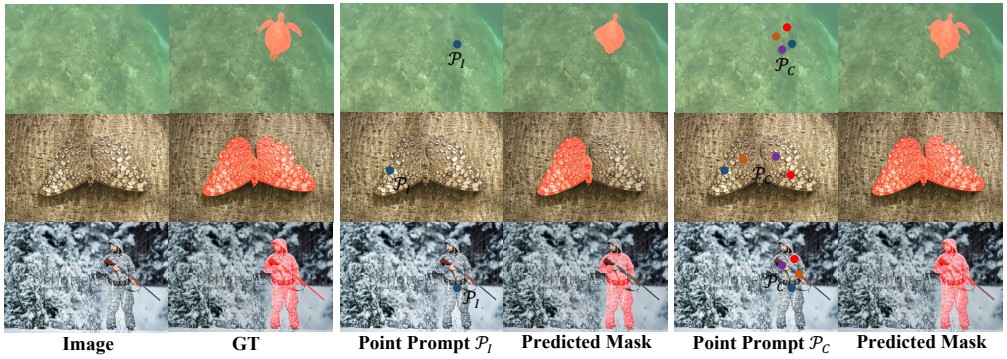

**Figure 7: The comparison between generated masks when using $\mathcal{P}_I$ and $\mathcal{P}_C$ as prompt points.**

**Table 5: Hyperparameters setting of $k$ and $c$ in this paper.**

| Methods | CAMO (250 Images) | | | COD10K (2026 Images) | | | NC4K (4121 Images) | | |
|---|---|---|---|---|---|---|---|---|---|
| | $F_\beta^w \uparrow$ | $S_\alpha \uparrow$ | MAE$\downarrow$ | $F_\beta^w \uparrow$ | $S_\alpha \uparrow$ | MAE$\downarrow$ | $F_\beta^w \uparrow$ | $S_\alpha \uparrow$ | MAE$\downarrow$ |
| Ours(Without VC) | 0.635 | 0.707 | 0.118 | 0.558 | 0.701 | 0.081 | 0.639 | 0.737 | 0.105 |
| Ours(k=16,c=4) | 0.679 | 0.750 | 0.101 | 0.591 | 0.734 | 0.064 | 0.680 | 0.766 | 0.081 |
| **Ours(k=16,c=3)** | **0.680** | **0.749** | **0.100** | **0.592** | **0.733** | **0.065** | **0.681** | **0.768** | **0.082** |
| Ours(k=16,c=2) | 0.675 | 0.740 | 0.102 | 0.588 | 0.732 | 0.068 | 0.678 | 0.766 | 0.084 |
| Ours(k=16,c=1) | 0.670 | 0.745 | 0.105 | 0.585 | 0.728 | 0.070 | 0.676 | 0.764 | 0.087 |
| Ours(k=4,c=3) | 0.672 | 0.744 | 0.106 | 0.581 | 0.723 | 0.070 | 0.673 | 0.758 | 0.088 |
| Ours(k=8,c=3) | 0.676 | 0.747 | 0.102 | 0.588 | 0.729 | 0.066 | 0.677 | 0.762 | 0.084 |
| **Ours(k=16,c=3)** | **0.680** | **0.749** | **0.100** | **0.592** | **0.733** | **0.065** | **0.681** | **0.768** | **0.082** |
| Ours(k=32,c=3) | 0.678 | 0.751 | 0.099 | 0.590 | 0.735 | 0.064 | 0.681 | 0.767 | 0.081 |

row indicate that using vanilla text prompt alone is insufficient to enable MLLM to perceive camouflaged scenes effectively. Subsequently, we enhance the text descriptions by including attributes of the camouflaged objects, and the text prompt is *"This image may contain a camouflaged object whose shape, color, texture, pattern and movement closely resemble its surroundings, enabling it to blend in. Can you identify it and provide its precise location coordinates?"*. The results from the second and third rows demonstrate a further improvement. After that, considering the issue of polysemy in descriptions, we modify the text prompts as *"This image may contain a concealed object whose shape, color, texture, pattern and movement closely resemble its surroundings, enabling it to blend in. Can you identify it and provide its precise location coordinates?"*. Using these two types of prompts in tandem to cue the MLLM, we observe an additional enhancement in performance. Finally, we generate synonymous prompts based on the first two text types to further cue the MLLM, thereby improving performance. The diverse text prompt is *"This image may contain a camouflaged object whose shape, color, pattern, movement and texture bear little difference compared to its surroundings, enabling it to blend in. Please provide its precise location coordinates."*.

In MMCPF, we also implement visual completion to further enhance the MLLM's ability to perceive camouflaged objects. The results in the sixth row demonstrate that incorporating visual completion can lead to further performance improvements. Fig. 7 visually illustrates the effectiveness of visual completion, showcasing

how this component of our approach significantly aids in the accurate detection and delineation of camouflaged objects.

**Hyperparameters Setting.** In VC, we use the Top-k algorithm to select the Top-k most similar points, followed by the clustering algorithm to group these points into $c$ cluster centers. As shown in Table 5, the choice of different $k$ and $c$ can all help improve the performance. Finally, in this paper, we choose $k = 16$ and $c = 3$.

## 5 CONCLUSION

This study successfully demonstrates that the MLLM can be effectively adapted to the challenging realm of zero-shot COD through our novel MMCPF. Despite the misinterpretation issues and localization uncertainties associated with MLLM in processing camouflaged scenes, our proposed CoVP significantly mitigates these challenges. By enhancing MLLM's perception from both linguistic and visual perspectives, CoVP not only reduces misinterpretation but also improves the precision in locating camouflaged objects. The validation of MMCPF across four major COD datasets confirms its efficacy, indicating that MLLM's generalizability extends to complex and visually demanding scenarios. This research not only marks a pioneering step in MLLM application but also provides a valuable blueprint for future endeavors aiming to enhance the perceptual capabilities of MLLMs in specialized tasks, paving the way for broader and more effective applications in the vision-language processing.

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
