# OpenReview forum: "Chain of Visual Perception: Harnessing Multimodal Large Language Models for Zero-shot Camouflaged Object Detection"
_acmmm.org/ACMMM/2024/Conference — MM2024 Poster_

### Official Review · Reviewer_u14o · 2024-04-29

**Rating:** 3
**Confidence:** 4

**Summary:**

This paper introduces a multimodal camo-perceptive framework (MMCPF) with strong generalization ability aimed at handling zero-shot Camouflaged Object Detection (COD) by leveraging the powerful capabilities of MLLMs. To enhance the effectiveness of MLLMs in COD, a strategy of Chain of Visual Perception is proposed to leverage both linguistic and visual cues for improving the perceptual capabilities of MLLMs in camouflaged scenes.

**Strengths:**

1. The paper is well-written and easy understanding. The figures are clear and intuitive.
2. The proposed text prompt mechanism and visual perception enhancement are simple and reasonable. The proposed method combine the strong capability of several foundation models for improving zero-shot camouflaged object detection.
3. Experiments exhibit that the proposed method achieves SOTA performance on public benchmarks, demonstrating its superior generalization ability.

**Limitations:**

1. The main concern is lack of novelty. The proposed method seems like a simple and direct combination of existing foundation models. Benefiting from their great pretrained capability, it is not surprising that the proposed method achieves SOTA performance. That is to say, the paper lacks interesting insights concerning camouflaged object detection, but a direct application of MLLMs on this task.
2. I would like to see the inference cost of the proposed method and previous methods (e.g., ERRNet). With MLLM, SAM and DINOv2, the total computation cost is quite high.
3. In the experiments, I would like to see the results with different sizes of backbones (e.g., MLLM-13B and SAM with more parameters.) In other words, can the proposed method benefits from more powerful foundation models.

**Suitability:**

3

---

### Official Review · Reviewer_SL8a · 2024-05-22

**Rating:** 4
**Confidence:** 4

**Summary:**

A multimodal camouflage-perceptive framework, termed MMCRF, is proposed in this paper for the field of zero-shot Camouflaged Object Detection (COD). The authors introduce the Chain of Visual Perception (CoVP) to enhance the capabilities of MLMs in detecting camouflaged objects in scenes, considering both visual and linguistic perspectives. The results on four widely-used datasets are promising.

**Strengths:**

1. The proposed MMCPF demonstrates strong zero-shot capabilities in the COD task.
2. With the proposed CoVP, MMCRF outperforms certain fully supervised methods and even some state-of-the-art methods under weakly-supervised settings.

**Limitations:**

1. To the best of my knowledge, the localization ability of MLMs is unsatisfactory. Even with instruction data that includes positional information, fine-tuned models like Shikra achieves decent performance. The authors expect to enhance the MLLM's localization performance for camouflaged objects simply through textual prompts. I am very curious about the extent of the improvement in localization ability this could bring, and I hope the authors can provide more details.
2. The MLLM output coordinates are used as the prompts for SAM, and the authors use k-NN to search more precise points for  camouflaged objects. However, this requires the MLLM providing a set of reasonably good initial points. Moreover, SAM is actually quite sensitive to the selection of sampling points. What are the authors' insights on the impact of these initial points? This issue is related to question 1.
3. The author used DINOv2 to extract semantic features from images, and then use k-NN to obtain some semantically closer points to guide SAM. Therefore, one might question why not to adopt a single framework like SEEM, which inherently has the capability for interactive segmentation based on position and also possesses semantic understanding. What is the inference speed when applying three large models within the entire zero-shot framework?
4. The format of the reference list should be checked carefully.

**Suitability:**

3

---

### Official Review · Reviewer_cwm1 · 2024-05-24

**Rating:** 4
**Confidence:** 3

**Summary:**

In this paper, the authors introduce a novel multimodal camo-perceptive framework (MMCPF), aimed at handling zero-shot Camouflaged Object Detection (COD) by leveraging the powerful capabilities of Multimodal Large Language Models (MLLMs).

**Strengths:**

The method proposed in this paper significantly enhances the effectiveness of Zero-shot Camouflaged Object Detection.

**Limitations:**

1）In this paper, the use of CoT technology to enhance the effectiveness of large language model prompts, while improving the overall model performance, is limited in terms of innovation in the CoT-based methods presented in this paper.

2）In Section 3.2.2, the authors utilize the DINOv2's  and KNN to expand upon the initial point coordinates. Will the stability of the aforementioned methods lead to fluctuations in the output of the detection model?

3）The experimental section lacks evaluation and analysis of model efficiency.

4）The shortcomings of this paper should be analyzed and discussed in detail

**Suitability:**

2

---

### Official Review · Reviewer_tjSf · 2024-05-25

**Rating:** 4
**Confidence:** 2

**Summary:**

This paper introduces a novel multimodal camo-perceptive framework (MMCPF) aimed at handling zero-shot Camouflaged Object Detection (COD) by leveraging the powerful capabilities of Multimodal Large Language Models (MLLMs). It identifies the limitations of current COD methods, which predominantly rely on supervised learning requiring extensive and accurately annotated datasets, leading to weak generalization. In response, the authors propose a zero-shot approach with the MMCPF that avoids these issues. A significant contribution of this work is the introduction of a mechanism called the Chain of Visual Perception (CoVP), which enhances the perceptual capabilities of MLLMs in camouflaged scenes by more effectively leveraging both linguistic and visual cues. The framework is tested on four COD datasets, and the results show that MMCPF outperforms existing zero-shot COD methods and achieves competitive performance compared to weakly-supervised and fully-supervised methods.

**Strengths:**

Innovative Framework: The introduction of the Chain of Visual Perception (CoVP) aims to enhance the model's understanding of complex camouflaged scenes. This approach is theoretically innovative, offering new insights for solving complex visual tasks.
The MMCPF leverages Multimodal Large Language Models (MLLMs) and does not require extensively annotated datasets, which is particularly beneficial for scenarios where acquiring sufficient labeled samples is challenging.

Generalization Capabilities: By not depending on specific training data, MMCPF is capable of performing better in unseen camouflage types and new environments, demonstrating enhanced generalization.

Writing： This paper is well-written.

**Limitations:**

It seems that there is a lack of data comparison on the CHAMELEON dataset, and Figure 5 lacks descriptions of the axes.

**I would like to add that I am not a specialist in COD and lack domain-specific knowledge, so I will refer to the opinions of other reviewers. Please do not worry; your work appears to be very solid. If other reviewers approve, I will consider raising my score.** : )

**Suitability:**

3

---

### Meta-Review · Area_Chair_h7cS · 2024-07-07

**Recommendation:** Accept (Poster)
**Confidence:** 4

**Metareview:**

This paper proposes to explore the capabilities of MLLMs for zero-shot Camouflaged Object Detection (COD). Specifically, the authors introduce the chain of visual perception to enhance MLLMs for COD. The reviewers consider the method is novel and are convinced of its effectiveness. After the rebuttal, most of the initial concerns from the reviewers are well addressed, and the paper receives 2 borderline accept and 1 weak accept.